# A Distributional Perspective on Pearl's Causal Hierarchy: From Marginal to Joint and Individualized Potential Outcomes

**Abstract**

Pearl's causal hierarchy has garnered sustained attention as a foundational lens for formulating and understanding causal questions, and has been extensively discussed within the framework of structural causal models. In this paper, we revisit the hierarchy from a potential outcomes perspective and propose a distributional interpretation of the causal hierarchy. We provide a formal and systematic classification of how different causal estimands correspond to distinct layers of the hierarchy. Building on this classification, we summarize key identifiability challenges for estimands at different layers and review general strategies for achieving identification under varying assumptions. Our perspective is both intuitive and theoretically grounded: higher layers of the hierarchy correspond to progressively richer probabilistic features of the potential outcomes distribution, which in turn require increasingly stronger assumptions for identification. We expect this perspective to help clarify and deepen understanding of various causal estimands, particularly those in the third layer of the causal hierarchy, along with their associated identifiability challenges, identifiability strategies, and application scenarios.

**Keywords:** Causal Inference, Ladder of Causation, Potential Outcomes

**Mathematics Subject Classification (2020):** 62D20

## 1 Introduction

Understanding causal relationships is a fundamental goal across a wide range of domains and has gained increasing attention in both academic and industry communities in recent years (Imbens and Rubin, 2015; Pearl, 2019; Hernán and Robins, 2020). Pearl articulates a three-layer causal hierarchy (Shpitser and Pearl, 2008; Pearl, 2009; Pearl and Mackenzie, 2018)–commonly referred to as the "Ladder of Causation"–that organizes causal reasoning into association, intervention, and counterfactuals (See Table 1 for details). This hierarchy distinguishes classes of causal queries with progressively greater conceptual and inferential demands, and has become a foundational lens for understanding the scope of causal analysis (Bareinboim et al., 2022). In addition, several studies have examined this hierarchy from complementary perspectives grounded in the structural causal model (SCM) framework, including logical-probabilistic, inferential-graphical, and computational complexity viewpoints (Bareinboim et al., 2022; Dörfler et al., 2025).

**Motivation.** Despite these important discussions, to the best of our knowledge, no work has formally examined Pearl's causal hierarchy from a potential outcomes lens. In this paper, we adopt this perspective to systematically address the question below.

Table 1: Pearl's causal hierachy ([Pearl](#), 2019). Questions at layer $l$ ($l = 1, 2, 3$) can be answered only if information from layer $l$ or higher is available. Here, $A$ denotes the treatment (intervention), $X$ the covariates, and $Y$ the outcome.

| Layer | Typical Activity | Typical Questions |
|---|---|---|
| 1. Association $\mathbb{P}(Y \mid X)$ | Seeing | What is? How would seeing $A$ change my belief in $Y$? |
| 2. Intervention $\mathbb{P}(Y \mid do(A = a), X)$ | Doing. Intervening | What if? What if I do $A$. |
| 3. Counterfactuals $\mathbb{P}(Y(a) \mid A = a', Y(a'))$ | Imagining. Retrospection | Why? Was it $A$ that caused $Y$? what if I had acted differently? |

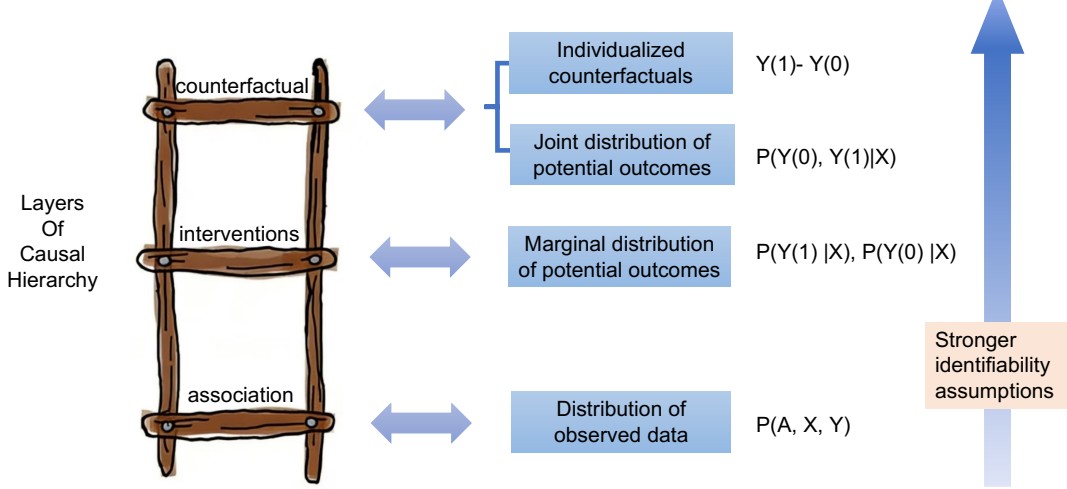

Figure 1: The proposed potential outcomes perspective on Pearl's causal hierarchy.

> Q1: How can we determine which layer of causation a given causal estimand belongs to, and how can diverse causal estimands be systematically classified?

**Our Perspective.** We address Q1 from the perspective of the probability distribution of potential outcomes. This perspective is fundamental and natural to statistical thinking and has been implicitly adopted across the causal inference literature. We formalize this perspective in a systematic manner. Our answer to Q1 is as follows:

> **Potential Outcomes Perspective**
> **Second layer: Intervention**. A causal estimand that depends only on the marginal distributions of potential outcomes (conditional on observed treatment and covariates) belongs to the second layer of causation.
> **Third layer: Counterfactuals**. A causal estimand that further depends on the joint distribution of potential outcomes (conditional on observed treatment and covariates), involves nested potential outcomes under different, mutually exclusive interventions (a.k.a. cross-world relationships), or targets potential outcomes at the individual level belongs to the third layer of causation.

We omit discussion of the first layer of causation, as causal inference primarily focuses on the second and third layers ([Pearl](#), 2019). Figure 1 illustrates the connection between the causal hierarchy and potential outcomes; see Section 3.1 for details. This perspective naturally aligns increasing layers of the hierarchy with progressively stronger identifiability assumptions–from marginal distributions to joint distributions and ultimately to individualized potential outcomes–

mirroring the inherent structure of the causal hierarchy.

**Contributions.** The main contributions are summarized as follows. First, we develop a probability distribution–based perspective on the layers of the causal hierarchy within the potential outcomes framework, and demonstrate its broad applicability. This perspective establishes a principled connection between the SCM and potential outcomes frameworks. Second, building on this perspective, we provide a systematic classification of a wide range of causal estimands, with a particular focus on those in the third layer and their applications; see Table 2 and Examples 1–11 for details. This classification enables a clearer formulation of causal problems, thereby delineating the scope and conditions under which a given method is valid. For example, when we aim to inform individual-level decision making for a specific unit, formulating the problem in terms of (conditional) average treatment effects at the second layer may lead to misleading conclusions (Lei and Candès, 2021; Kallus, 2022). Third, based on this classification, we summarize the key identifiability challenges associated with causal estimands at each layer and review the corresponding identifiability strategies; see Table 2 for details. This helps determine, for a given causal estimand, whether the imposed identifiability assumptions are sufficient or overly restrictive, and delineates the boundary conditions under which specific methods can appropriately address the causal questions at hand.

The proposed perspective also reveals a general organizing principle underlying Pearl's causal hierarchy: each ascent in the hierarchy requires identifying a progressively richer probabilistic object. Association-level reasoning concerns the observed data distribution; intervention-level reasoning requires identifying the marginal distributions of potential outcomes; cross-world causal queries further require identifying the joint distributions of potential outcomes; and individualized counterfactual reasoning ultimately concerns individual-level realizations of potential outcomes. This progression naturally explains why higher layers of the hierarchy demand stronger identifiability assumptions and increasingly structured modeling assumptions.

In summary, our discussion enables researchers to quickly assess, for a given scientific problem, whether a particular causal estimand is well aligned with the scientific question of interest, and, for a given causal estimand, whether the associated identifiability assumptions are sufficient or overly restrictive. When these assumptions are insufficient or overly restrictive, our framework further suggests strategies for improving identification. In this sense, the discussion presented in this paper, summarized in Figure 1 and Table 2, provides a useful complement to existing applications of the potential outcomes framework.

## 2 Preliminaries

### 2.1 Notation

Let $A \in \mathcal{A}$ denote the treatment (or intervention), $X \in \mathcal{X}$ a vector of pre-treatment covariates, and $Y \in \mathcal{Y}$ an outcome variable. Under the potential outcomes framework, let $Y(a)$ denote the potential outcome that would be observed if the treatment were set to $A = a$. Under the stable unit treatment value assumption, i.e., there are no multiple versions of treatment and no interference among individuals, the observed outcome is $Y = Y(A)$. Let $\mathbb{P}$ denote the target population of interest, with $\mathbb{E}$ the corresponding expectation operator.

With this notation in place, we distinguish between factual and counterfactual outcomes. Potential outcomes are defined prior to measurement; after measurement, one potential outcome becomes factual while the others remain counterfactual (Wang et al., 2025b). Specifically, for an individual with $A = a$, $Y(a)$ is the factual (observed) outcome, whereas $Y(a')$ for $a' \neq a$ is the counterfactual (unobserved) outcome; see Figure 2 for illustration.

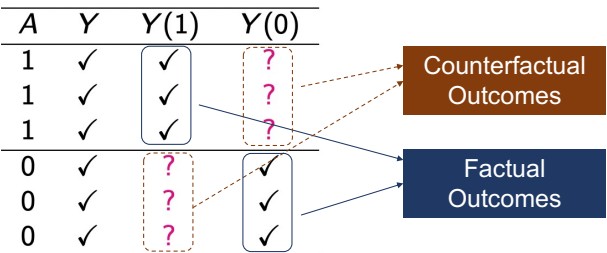

Figure 2: Illustration of factual and counterfactual outcomes under binary treatment, where ✓ and ? mean observed and unobserved, respectively. We omit $X$ for simplicity.

In Section 3.5, we introduce a post-treatment variable observed after treatment assignment and before the outcome. This facilitates discussion of long-term causal effects, mediation analysis, counterfactual fairness metrics, and principal causal effects, demonstrating the applicability of our proposed perspective and classification.

## 2.2  Pearl's Causal Hierarchy and Structural Causal Model

We provide a brief introduction to Pearl's causal hierarchy and the definition of the structural causal model.

**Pearl's Causal Hierarchy**. We present Pearl's causal hierarchy in Table 1 and highlight two key points. First, interventional queries (the second layer) are typically prospective, as they reason about the effects of hypothetical interventions on future outcomes based on current observations, whereas counterfactual queries (the third layer) are retrospective, as they consider alternative past outcomes for events that have already occurred (Pearl, 2019). Second, questions at layer $l$ ($l = 1, 2, 3$) can be answered only if information from layer $l'$ with $l' \geq l$ is available Ibeling and Icard (2020); Bareinboim et al. (2022). See Pearl (2019) for more details.

**Structural Causal Model** (SCM, Pearl, 2009). An SCM $\mathcal{M}$ consists of a graph $\mathcal{G}$ and a corresponding set of structural equation models $\mathcal{F} = \{f_1, ..., f_p\}$. The nodes in $\mathcal{G}$ are divided into two categories: (a) exogenous variables $\mathbf{U} = (U_1, ..., U_p)$, which represent the environment during data generation, assumed to be mutually independent; (b) endogenous variables $\mathbf{V} = \{V_1, ..., V_p\}$, which denote the relevant features that we need to model in a question of interest. For variable $V_j$, its value is determined by a structural equation $V_j = f_j(PA_j, U_j)$, $j = 1, ..., p$, where $PA_j$ stands for the set of parents of $V_j$. The SCM provides a formal language for describing how the variables interact and how the resulting distribution would change in response to certain interventions.

The SCM will be used to characterize counterfactual outcomes in Section 3.4. For a detailed discussion of how SCMs can be formulated in terms of potential outcomes and how the two

frameworks can be translated into one another, we refer the reader to Richardson and Robins (2013) and Wang et al. (2025b).

# 3 A Potential Outcomes Perspective

In this section, we provide a detailed exposition of the potential outcomes perspective on Pearl's causal hierarchy. Section 3.1 presents an overview, while Sections 3.2–3.4 focus on the second and third layers of causation, offering a classification of various causal estimands and highlighting key identifiability challenges and associated identifiability strategies across layers. Section 3.5 extends these discussions to settings involving post-treatment variables.

## 3.1 Overview

Figure 1 illustrates the proposed potential outcomes perspective and its correspondence with Pearl's causal hierarchy. We briefly describe this perspective below.

**Second layer: Intervention.** Causal estimands at this layer are functionals of the marginal distributions of potential outcomes. In other words, each potential outcome corresponds to the outcome in an intervention world in which all individuals receive the same treatment, and second-layer estimands depend only on one or more marginal potential outcome distributions under a single intervention. A causal query (or estimand) that can be decomposed into one or more sub-queries, each involving a single intervention, belongs to the second layer. Treatment randomization identifies the marginal distributions of potential outcomes and is therefore sufficient to identify estimands at this layer. Typical causal estimands at this level include the average treatment effect, the quantile treatment effect, and the dose-response function. See Section 3.2 and Table 2 for details.

**Third layer: Counterfactuals.** We divide this layer into two sublayers by inferential complexity.

*Sublayer 1: Cross-World Causal Queries.* Causal estimands at this layer depend on features of the joint distribution of potential outcomes, or equivalently on nested potential outcomes defined under multiple, mutually exclusive interventions. Such estimands cannot be expressed as functionals of marginal potential outcome distributions alone. Because they simultaneously reference outcomes from different interventional worlds, we refer to these estimands as cross-world causal queries. Treatment randomization alone identifies only marginal potential outcome distributions and is therefore insufficient for identifying estimands at this layer. Typical estimands at this layer include attribution metrics, benefit and harm rates, persuasion rate, and natural direct and indirect effects. Estimands at this layer often provide complementary information to those at the second layer, supporting more nuanced treatment selection and evaluation. See Section 3.3 and Table 2 for more details.

*Sublayer 2: Individual-Level Counterfactual Queries.* This layer focuses on individualized treatment effects (ITEs). Learning ITEs requires much stronger identifiability assumptions than those needed for estimands in other layers.

Table 2: Classification of various estimands and their associated identifiability strategies. Note: this table lists only a subset of typical causal estimands and is not intended to be exhaustive.

| Layer | Typical Estimands (or Classes Thereof) | Main Challenges | (Partial) Identifiability Strategies |
|---|---|---|---|
| Second Layer (Marginal Distribution) | Average Treatment Effect, Causal Risk and Odd Ratios (Example 1) Quantile and Distributional Treatment Effects (Example 1) Dose-Response Function, Average Derivative Effect (Example 2) Short-Term and Long-Term Treatment Effects (Example 9) Counterfactual Parity and Total Effect (Examples 10 and 11) | (Unmeasured) Confounding Between Treatment and Outcome | Ignorability (or Randomization) Auxiliary Variable (e.g., IV) Data Fusion Sensitivity Analysis |
| Third Layer (Joint Distribution or Nested Potential Outcomes in Cross Worlds) | Probability of Causation (Example 3) Treatment Benefit and Harm Rates (Example 4) Effect of Persuasion (Example 5) Distribution of the ITE (Example 6) Principal Causal Effects (Example 8) Principal Fairness (Example 10) Natural Direct and Indirect Effects (Example 11) | (Unmeasured) Confounding Between Treatment and Outcome + Dependence Between Cross-World Potential Outcomes | Ignorability + Independence Ignorability + Monotonicity Ignorability + Association Parameter Ignorability + Copula Model Data Fusion Sensitivity Analysis Under Ignorability Partial Identification Under Ignorability |
| Third layer (Individual Level) | Individualized Treatment Effect Counterfactual Fairness (Example 10) | Connection Between Factual and Counterfactual Outcomes at the Individual Level | Abduction-Action-Prediction Rank Preservation Conformal Inference |

Note: The notation "$A + B$" denotes $A$ and $B$; that is, both $A$ and $B$ are required or exist.

**Remark 1** *The above perspective and classification exhibit a clear monotonic relationship: knowing individual-level counterfactual outcomes implies knowledge of the joint distribution of potential outcomes, and knowing the joint distribution implies knowledge of the marginal distributions–but not vice versa. This monotonicity aligns with the inherent progression across the layers of Pearl's causal hierarchy.*

In this paper, we primarily focus on the third layer consisting of two sublayers, and only briefly discuss the second layer, as it has been extensively studied in the existing literature and textbooks (Imbens and Rubin, 2015; Rosenbaum, 2020; Hernán and Robins, 2020; Ding, 2023). In Sections 3.2–3.5 below, we omit covariates $X$ unless explicitly stated.

## 3.2 Second Layer: Intervention

We introduce typical causal estimands at the second layer of causation.

**Example 1 (Binary Treatment, Second Layer)**
- *Average treatment effect (ATE):* $\text{ATE} = \mathbb{E}[Y(1) - Y(0)]$.
- *ATE on treated (ATT):* $\text{ATT} = \mathbb{E}[Y(1) - Y(0) \mid A = 1]$.
- *Quantile treatment effect (QTE, Firpo, 2007):* $\text{QTE}_\tau = q_{1,\tau} - q_{0,\tau}$, where $q_{a,\tau} = \inf\{q : \mathbb{P}(Y(a) \leq q) \geq \tau\}$ is the $\tau$-quantile of the distribution of $Y(a)$ for $a = 0, 1$.
- *Distributional treatment effect (DTE, Oka et al., 2025):* $\text{DTE}(y) = F_{Y(1)}(y) - F_{Y(0)}(y)$, where $F_{Y(a)}(\cdot)$ is the cumulative distribution of $Y(a)$ for $a = 0, 1$, and $y \in \mathcal{Y}$.
- *In addition, for binary outcomes, we usually define causal risk ratio (CRR) and causal odd ratio (COR):*
$$\text{CRR} = \frac{\mathbb{P}(Y(1) = 1)}{\mathbb{P}(Y(0) = 1)}, \quad \text{COR} = \frac{\mathbb{P}(Y(1) = 1)/\mathbb{P}(Y(1) = 0)}{\mathbb{P}(Y(0) = 1)/\mathbb{P}(Y(0) = 0)}.$$

**Example 2 (Continuous Treatment, Second Layer)**
- *Effect curve or dose–response function (DRF) (Galvao and Wang, 2015; Kennedy et al., 2017):* $\text{DRF}(a) = \mathbb{E}[Y(a)]$.
- *Average derivative effect (ADE, Newey and Stoker, 1993; Dong et al., 2025):*

$$\text{ADE}(a) = \frac{\partial \mathbb{E}[Y(a)]}{\partial a}.$$

For identifying estimands at the second layer, the primary challenge arises from confounding between treatment and outcomes. The most commonly used identifiability conditions are ignorability ($A \perp\!\!\!\perp Y(a)$), also referred to as no unmeasured confounding (Rosenbaum and Rubin, 1983), and the overlap condition ($0 < \Pr(A = 1) < 1$), which is satisfied under a well-designed randomized treatment assignment. When randomization is infeasible and unmeasured confounding is a concern, researchers may pursue alternative strategies. These include leveraging auxiliary variables, such as instrumental variables (Imbens, 2004; Wang and Tchetgen Tchetgen, 2018) and negative controls (Lipsitch et al., 2010; Hu et al., 2023); adopting data-fusion approaches that combine multiple complementary data sources (Colnet et al., 2024; Wu et al., 2025b); and exploiting structural restrictions in multi-dimensional treatments and/or outcomes (Zhou et al., 2024; Tang et al., 2026). In addition, sensitivity analysis provides a fundamental tool for assessing the robustness of causal conclusions to violations of ignorability (Kallus and Zhou, 2018; Rosenbaum, 2020; Ding et al., 2022; Ding, 2023).

## 3.3 Third Layer: Cross-World Causal Queries

In this subsection, we first present various causal estimands involving the joint distribution of potential outcomes, and then discuss the key identifiability challenges and review the associated identification strategies.

### 3.3.1 Causal Estimands

We present representative third-layer estimands involving joint distributions in the following examples, each illustrating an interesting application scenario.

**Example 3 (Probability of Causation)** *Causal inference concerns not only the effects of causes but also the causes of observed effects (Dawid and Musio, 2022), the latter often referred to as attribution analysis (Pearl, 2009; Pearl et al., 2016). For binary treatment and outcomes, two standard attribution measures are the probability of necessary causation (PN) and the probability of sufficient causation (PS), defined as follows (Pearl, 1999; Tian and Wu, 2025):*

$$\mathrm{PN}(A \Rightarrow Y) = \mathbb{P}(Y(0) = 0 \mid A = 1, Y = 1),$$
$$\mathrm{PS}(A \Rightarrow Y) = \mathbb{P}(Y(1) = 1 \mid A = 0, Y = 0).$$

*The PN provides a probabilistic formalization of the but-for principle in legal reasoning, while PS captures a complementary notion of causal sufficiency (Pearl and Mackenzie, 2018). In recent years, several studies have extended PN and PS to non-binary treatments, multiple binary treatments, and non-binary outcomes; see Lu et al. (2023); Zhao et al. (2023); Li and Pearl (2024); Li et al. (2024); Zhang et al. (2025); Luo et al. (2025) for more details.*

**Example 4 (Treatment Benefit and Harm Rates)** *For a binary treatment, assuming larger outcomes are preferable, the treatment harm rate (THR) and treatment benefit rate (TBR) (Gadbury et al., 2004; Zhang et al., 2013; Huang et al., 2012; Sarvet and Stensrud, 2023; Mueller and Pearl, 2023; Shen et al., 2013; Yin et al., 2018; Kallus, 2022; Li et al., 2023; Wu et al.,*

$$\text{THR} = \mathbb{P}(Y(1) - Y(0) < 0),$$
$$\text{TBR} = \mathbb{P}(Y(1) - Y(0) > 0).$$

*These quantities measure the proportions of individuals who experience better or worse outcomes under treatment compared to control, respectively. In addition, one may define the treatment harm quantity (THQ) as*

$$\text{THQ} = \mathbb{E}[(Y(0) - Y(1))\mathbb{I}(Y(1) - Y(0) < 0)],$$

*which quantifies the magnitude of treatment-induced harm. The THR, TBR, and THQ provide complementary insights into individual-level treatment effect heterogeneity that are not captured by the (conditional) ATE, offering crucial guidance for individualized decision-making.*

**Example 5 (Effect of Persuasion)** *Let $A \in \{0,1\}$ denote a binary indicator of an individual's exposure to persuasive information, and let $Y$ be a binary outcome, where $Y = 0$ indicates a negative response and $Y = 1$ indicates a positive response. The persuasion rate (Jun and Lee, 2023, 2024) is defined as*

$$\text{PR} = \mathbb{P}(Y(1) = 1 \mid Y(0) = 0),$$

*which quantifies the proportion of individuals who would respond negatively in the absence of exposure but whose behavior becomes positive after exposure to persuasive information.*

**Example 6 (Distribution of ITE)** *For binary treatments, we can define the distribution of the ITE as $F(\delta) = \mathbb{P}(Y(1) - Y(0) \leq \delta)$, as studied by Kim (2014); Yin et al. (2018); Shin (2025). This distribution differs from the $\text{DTE}(y)$ introduced in Example 1. Several studies have considered variants of $F(\delta)$. For example, Kallus (2023) defined the conditional value at risk (CVaR) of the ITE distribution,*

$$\text{CVaR}_\alpha = \mathbb{E}[Y^1 - Y^0 \mid Y^1 - Y^0 \leq q_\alpha],$$

*where $q_\alpha$ is the $\alpha$-quantile of $Y^1 - Y^0$. The $\text{CVaR}_\alpha$ measures the average treatment effect in the lower $\alpha$-quantile of the treatment effect distribution. Kaji and Cao (2025) studied estimands of the form $\mathbb{E}[Y(1) - Y(0) \mid Y(0) \leq c]$. If $Y$ denotes wealth, this represents the average treatment effect for the subgroup that would have low wealth if untreated.*

### 3.3.2 Identifiability Challenges and Corresponding Strategies

Identifying causal estimands at this layer requires characterizing the joint distribution of potential outcomes. Because both potential outcomes are never observed simultaneously for any individual, this task poses substantial identification challenges. Beyond confounding between treatment and outcome, one must additionally characterize the dependence structure between potential outcomes. Randomization resolves the former, but not the latter, and therefore does not generally identify third-layer estimands. Below, we review various strategies for addressing this dependence in the case of a binary treatment.

- *Independence Between Potential Outcomes:* $Y(1) \perp\!\!\!\perp Y(0)$. Under this assumption, the joint distribution of potential outcomes $(Y(1), Y(0))$ is determined by their marginal distributions. Yin et al. (2018) relax this assumption to latent independence, $Y(1) \perp\!\!\!\perp Y(0) \mid U$ for a latent variable $U$, at the cost of additional parametric model restrictions.

- *Monotonicity Between Potential Outcomes:* $Y(1) \geq Y(0)$ *a.s.*. Monotonicity addresses the dependence between $Y(1)$ and $Y(0)$ for binary outcomes and has been widely used in the literature; see, e.g., Tian and Pearl (2000); Cai and Kuroki (2005); Huang et al. (2012). The implication of this condition is straightforward: under monotonicity and ignorability, the joint distribution $\mathbb{P}(Y(1), Y(0))$ involves three unknown parameters, $\pi_{jk} = \mathbb{P}(Y(1) = j, Y(0) = k)$ for $j, k \in 0, 1$ with $\pi_{01} = 0$, which are identified by the system of three equations: $\pi_{10} + \pi_{11} = \mathbb{P}(Y(1) = 1), \pi_{01} + \pi_{11} = \mathbb{P}(Y(0) = 1), \sum_{j=0}^{1} \sum_{k=0}^{1} \pi_{jk} = 1$. Nevertheless, an interesting implication of the monotonicity assumption is an asymmetry in its identifying power. While monotonicity is sufficient to identify the joint distribution of potential outcomes when the treatment takes multiple ordered levels and the outcome is binary (Wang et al., 2017a), it generally does not yield point identification when outcomes are ordinal or continuous, even with binary treatment (Zhang et al., 2025; Lu et al., 2025; Zhang and Yang, 2025).

- *Specification of Association Parameter Between Potential Outcomes.* Under the ignorability and overlap assumptions, for binary outcomes, the joint distribution of $(Y(1), Y(0))$ is identifiable for a given Pearson correlation coefficient $\rho = \mathrm{Corr}(Y(1), Y(0))$ (Wu et al., 2026), or a given odds ratio (Ciocănea-Teodorescu et al., 2025; Tong et al., 2025)

$$\mathrm{OR} = \frac{\mathbb{P}(Y(1) = 1 \mid Y(0) = 1)\mathbb{P}(Y(1) = 0 \mid Y(0) = 0)}{\mathbb{P}(Y(1) = 0 \mid Y(0) = 1)\mathbb{P}(Y(1) = 1 \mid Y(0) = 0)}.$$

When specifying a single association parameter is difficult, one may instead posit a plausible range for it and derive corresponding lower and upper bounds for the joint distribution. This naturally yields a sensitivity analysis framework, with the association parameter serving as the sensitivity parameter. A key insight of Wu et al. (2026) is that assuming a positive association parameter is often reasonable in real-world applications and can substantially tighten the bounds on the joint distribution of potential outcomes.

- *Copula Models for Continuous Outcomes.* Specifying an association parameter alone is insufficient to identify the joint distribution of potential outcomes for continuous outcomes, necessitating additional model restrictions. Copula models offer a classical framework for constructing joint distributions from marginal distributions by explicitly modeling their dependence structure (Jaworski et al., 2010; Bartolucci and Grilli, 2011; Sun et al., 2024; Lu et al., 2025; Zhang and Yang, 2025). For example, a Gaussian copula assumes that $(Y(1), Y(0))$ follows a joint Gaussian distribution with a given correlation coefficient; varying this coefficient naturally yields a sensitivity analysis framework.

- *Data Fusion.* For binary and categorical outcomes, Wu and Mao (2026) establishes nonparametric identification of the joint distribution under binary treatment by combining multiple experimental studies, and Shahn and Madigan (2025) subsequently extends this framework.

Although researchers can use the above strategies to address the identification of the joint distribution of potential outcomes, the required conditions may be too restrictive in practice. Consequently, partial identification naturally arises as a central framework for third-layer causal

inference. Rather than aiming for full recovery of the joint distribution of potential outcomes, partial identification methods characterize the set of distributions or provide bounds on the joint distribution of potential outcomes, compatible with the observed data and the maintained assumptions, see e.g., Fan and Park (2009, 2010); Fan et al. (2014); Kim (2014); Firpo and Ridder (2019); Frandsen and Lefgren (2021); Kaji and Cao (2025).

## 3.4 Third Layer: Individual-Level Counterfactual Queries

Different from Sections 3.2–3.3, this layer first requires clarifying the well-definedness of the task of learning individual-level counterfactual outcomes.

### 3.4.1 Well-Definedness and Identifiability Challenges

From a probabilistic perspective, learning individual-level counterfactual outcomes is generally infeasible, as they are random variables rather than well-defined causal estimands. To address this issue, we need to impose the assumption below.

**Assumption 1 (Deterministic Counterfactual Outcomes, Pearl et al., 2016)** *For the population of interest, we treat each individual's potential outcome $Y(a)$ as a fixed quantity.*

Assumption 1 adopts a deterministic view of counterfactual outcomes, which differs from the stochastic perspective underlying statistical modeling and inference, where $Y(a)$ is treated as a random variable Dawid (2000). This deterministic view is adopted primarily as an operational device for reasoning about individualized counterfactual outcomes. Under Assumption 1, learning individual-level counterfactual outcomes becomes feasible under suitable conditions. Beyond confounding between treatment and outcome, the key challenge lies in establishing a connection between the factual and counterfactual outcomes at the individual level. Before presenting the identifiability strategies, we provide an example illustrating the risk of individual decision-making based on the conditional ATE (CATE), an estimand in the second layer.

**Example 7 (Risks of Decision-Making Based on CATE)** *In policy learning and personalized medicine, it is common to recommend treatment for an individual with $X = x$ if their CATE, $\tau(x) = \mathbb{E}[Y(1) - Y(0) \mid X = x]$, is greater than zero. However, $\tau(x)$ is an average metric over the subpopulation with $X = x$, and using it for individualized decision-making can be misleading (Ding et al., 2019; Lei and Candès, 2021; Mueller and Pearl, 2023). For example, consider a subpopulation with covariates $X = x$ consists of 10 individuals, and $\tau(x) = 0.1$. Based on CATE, we would recommend treatment for all individuals in this subpopulation. However, it may be the case that five individuals have ITEs of 1, while the other five have ITEs of -0.8. Recommending treatment for all 10 individuals would therefore harm half of the subpopulation. To address this problem and promote safer decision-making, we can use the harm rate as a complementary criterion for treatment recommendations (Li et al., 2023; Kallus, 2022; Wu et al., 2026), or construct prediction intervals for the ITE (Jin et al., 2023).*

### 3.4.2 Identifiability and Estimation Strategies

Pearl et al. (2016) proposed the widely used three-step procedure–abduction, action, and prediction–for estimating counterfactual outcomes, briefly summarized below.

**Pearl's Three-Step Procedure.** Suppose we have three sets of variables $A$, $Y$, and $\mathbf{E} \subseteq \mathbf{V}$ in a structural causal model (SCM) $\mathcal{M}$. Consider the counterfactual query: for an individual with observed evidence $\mathbf{E} = \mathbf{e}$, what would have happened had we set $A$ to a different value $a'$? Pearl's three-step procedure addresses this question as follows: (a) **Abduction**: infer the value of the exogenous variables $\mathbf{U}$ for this individual given the evidence $\mathbf{E} = \mathbf{e}$; (b) **Action**: modify $\mathcal{M}$ by replacing the structural equations for $A$ with $A = a'$, yielding the intervened model $\mathcal{M}_{a'}$; (c) **Prediction**: use $\mathcal{M}_{a'}$ together with the inferred $\mathbf{U}$ to compute the counterfactual outcome of $Y$ for this individual.

For clarity, we present an example illustrating the application of Pearl's three-step procedure. Specifically, let $\mathbf{V} = (X, A, Y)$, where $A$ causes $Y$, $X$ affects both $A$ and $Y$, and the structural equation of $Y$ is given as

$$Y = f_Y(A, X, U). \tag{1}$$

Let $Y(a')$ denote the potential outcome that would result if $A$ were set to $a'$. The counterfactual question,"given the observed evidence $(A = a, X = x, Y = y)$ for an individual, what would have happened had $A$ been set to $a'$?", corresponds to estimating $y_{a'}$, the realized value of $Y(a')$ for that individual. Using Pearl's three-step procedure, estimation of $y_{a'}$ proceeds as follows: (a) *Abduction*: infer the individual-specific value of the exogenous variables, denoted by $u$; (b) *Action*: intervene to set $A = a'$; and (c) *Prediction*: compute the counterfactual outcome as $f_Y(a', x, u)$.

**Analysis of Pearl's Three-Step Procedure.** Pearl's three-step procedure provides an elegant and easily manipulable method for estimating individual-level counterfactual outcomes. However, its application requires two prerequisites: (a) The availability of an SCM that describes the data-generating process (Brouwer, 2022; Xie et al., 2023); and (b) the exogenous variables must be identifiable or estimable from the SCM. For instance, in model (1), one needs to determine the functional form of $f_Y$ and obtain the value of $U$ using the inverse of $f_Y$. These two prerequisites may limit the applicability of Pearl's three-step procedure.

We further summarize two alternative methods–quantile regression and conformal inference–for estimating and bounding individual-level counterfactual outcomes, respectively.

**Rank Preservation and Quantile Regression**. Xie et al. (2023) established the identifiability of counterfactual outcomes under ignorability and a strict monotonicity assumption, under which the outcome is a strictly monotone function of an exogenous variable. Under this assumption, for any individual, the factual and counterfactual outcomes share the same quantile in their respective marginal distributions of potential outcomes. For estimation, they proposed a quantile regression method to estimate counterfactual outcomes, thereby avoiding the need to specify or estimate a structural causal model. Wu et al. (2025a) further showed that the strict monotonicity assumption is a special case of rank preservation, under which an individual's factual and counterfactual outcomes share the same rank in the corresponding outcome distributions for all individuals (Heckman et al., 1997; Chernozhukov and Hansen, 2005), and

proposed an improved estimation method. Wang et al. (2025a) used a similar technique to estimate fairness metrics, with applications to reinforcement learning.

From a distributional perspective, rank preservation implicitly specifies a particular coupling between the factual and counterfactual outcome distributions that coincides with the optimal transport map, which for one-dimensional distributions aligns corresponding quantiles. This connection places rank preservation within the broader framework of optimal transport, as, for example, studied for causal effects on outcome distributions by Lin et al. (2023).

**Conformal Inference**. Instead of estimating counterfactual outcomes for each individual, conformal inference aims to construct prediction intervals for them (Lei and Candès, 2021; Jin et al., 2023; Bodik et al., 2025). For example, in the case of a binary treatment, conformal inference seeks to find prediction intervals $C_{\text{ITE}}(X)$ for the individual treatment effect (ITE) such that, for a given $\alpha \in (0,1)$,

$$\mathbb{P}(Y(1) - Y(0) \in C_{\text{ITE}}(X; \alpha)) \geq 1 - \alpha,$$

or $\mathbb{P}(Y(1) - Y(0) \in C_{\text{ITE}}(X; \alpha) \mid X = x) \geq 1 - \alpha$. For an individual $i$, without loss of generality, if $Y_i(0)$ is observed and $Y_i(1)$ is missing, we only need to construct prediction interval for $Y_i(1)$, i.e., find $C_1(X_i; \alpha)$ such that $\mathbb{P}(Y_i(1) \in C_1(X_i; \alpha)) \geq \alpha$. It is then natural to define $C_{\text{ITE}}(X_i; \alpha) = C_1(X_i; \alpha) - Y_i(0)$. Under the ignorability and overlap assumptions, constructing $C_1(X_i; \alpha)$ reduces to a standard conformal inference problem with covariate shift between $\mathbb{P}(X)$ and $\mathbb{P}(X \mid A = 1)$ (Lei and Candès, 2021). If both $Y_i(0)$ and $Y_i(1)$ are missing, we can first construct prediction intervals $C_0(X_i; \alpha/2)$ and $C_1(X_i; \alpha/2)$ for $Y_i(0)$ and $Y_i(1)$, respectively, and then define $C_{\text{ITE}}(X_i; \alpha) = \{z_1 - z_2 : z_1 \in C_1(X_i; \alpha/2),\ z_2 \in C_0(X_i; \alpha/2)\}$ (Jin et al., 2023).

The prediction intervals $C_{\text{ITE}}(X_i; \alpha)$ are often wide. Bodik et al. (2025) obtained narrower intervals by introducing a mild specification (positive correlation) of the association parameters between potential outcomes, extending the work of Wu et al. (2026).

**Comparison.** We compare the merits of rank preservation (quantile regression) and conformal inference methods, as summarized in Table 3.

Table 3: Comparison of rank preservation and conformal inference.

| Property | Rank Preservation | Conformal Inference |
|---|---|---|
| Weaker Conditions | ☹ | ☺ |
| Point Identification | ☺ | ☹ |
| Generalizable | ☹ | ☺ |

Estimating individual-level counterfactual outcomes yields point estimates of the ITE, but this typically relies on strong identifiability assumptions (e.g., rank preservation). Moreover, such approaches are inherently retrospective and difficult to generalize, since counterfactual estimation depends on observed factual outcomes and therefore applies only to populations with observed outcomes. In comparison, conformal inference provides a promising framework for conducting inference on the ITE with high probability while relying on substantially weaker assumptions. Moreover, it exhibits strong generalization capability, in that it can be applied to new populations for which only covariates are observed. Nevertheless, the resulting prediction

intervals are often wide and provide only limited information about the ITE, restricting their applicability.

## 3.5 In the Presence of Post-Treatment Variable

In this subsection, we adopt the proposed potential outcomes perspective to briefly analyze scenarios in which an additional post-treatment variable is present. In addition to the observed variables $(A, X, Y)$, suppose that we also observe a variable $S$ measured after the treatment $A$ and before the outcome $Y$. Consider a binary treatment, and let $S(0)$ and $S(1)$ denote the potential outcomes of $S$.

**Example 8 (Principal Causal Effects)** *The principal causal effect is defined as*

$$\mathbb{E}[Y(1) - Y(0) \mid S(1) = s1, S(0) = s0].$$

*This estimand belongs to the third layer, as it involves the joint distribution of $(S(0), S(1))$. The principal causal effect has many practical applications, including noncompliance (Imbens and Angrist, 1994), truncation by death (Rubin, 2006), and surrogate endpoint evaluation (Jiang et al., 2016; Wu and Mao, 2026). See Frangakis and Rubin (2002); Wang et al. (2017b); Lu et al. (2025); Zhang and Yang (2025) for further examples.*

**Example 9 (Short-term and Long-term Treatment Effects)** *Let $S$ and $Y$ denote the short-term and long-term outcomes, respectively. Then $\mathbb{E}[S(1) - S(0)]$ and $\mathbb{E}[Y(1) - Y(0)]$ represent the short-term and long-term treatment effects, respectively, which belong to estimands in the second layer. A key characteristic of this setting is that long-term outcomes are difficult to observe and often suffer from substantial missingness due to extended follow-up periods, drop-outs, and budget constraints (Athey et al., 2025; Chen and Ritzwoller, 2023; Kallus and Mao, 2025; Hu et al., 2025; Imbens et al., 2025). Beyond addressing confounding between treatment and both short- and long-term outcomes, one must also account for the missingness of long-term outcomes (Athey et al., 2025). Several studies investigate how to learn policies that maximize long-term outcomes (Yang et al., 2024a; Huang and Ascarza, 2024), or how to balance short- and long-term outcomes (Wu et al., 2024; Yang et al., 2024b).*

**Example 10 (Causal Fairness Metrics)** *Counterfactual fairness has gained increasing attention in recent years (Davide and Bradic, 2024; Li et al., 2025). Let $A$ denote a protected attribute (e.g., gender), $Y$ an outcome or task-relevant attribute (e.g., grade), and $S \in \{0, 1\}$ the decision (e.g., admission) made by a machine learning algorithm. The goal is to evaluate the fairness of the algorithm. Several counterfactual fairness metrics have been proposed:*

*• Counterfactual parity (Mitchell et al., 2021): $\mathbb{P}(S(1) = 1) = \mathbb{P}(S(0) = 1)$. It belongs to the second layer.*

*• Counterfactual fairness (Kusner et al., 2017): $S_i(1) = S_i(0)$ for all individual $i \in \{1, ..., N\}$. It belongs to the third layer (individual-level counterfactual outcomes).*

*• Principal fairness (Imai and Jiang, 2023; Li et al., 2025): $\mathbb{P}(S = 1 \mid Y(1), Y(0), A) = \mathbb{P}(S = 1 \mid Y(1), Y(0))$ and principal counterfactual parity: $\mathbb{P}(S(0) = 1 \mid Y(1) = Y(0)) =$*

$\mathbb{P}(S(1) = 1 \mid Y(1) = Y(0))$. *These quantities belong to the third layer, as they involve the joint distribution of potential outcomes.*

**Example 11 (Mediation Analysis)** *We denote the mediator variable by $M$ instead of $S$ by convention. We introduce the potential outcome $Y(a, m)$ corresponding to treatment $A = a$ and $M = m$. Pearl (2001) further considered the nested potential outcome $Y(a, M(a))$, which represents the hypothetical outcome if the treatment were set to level $a$ and the mediator were set to its potential level $M(a)$ under treatment $a$. The potential outcome $Y(a)$ is defined as $Y(a, M(a))$ for $a = 0, 1$ (referred to as the composition assumption; Ding, 2023).*

*• The total effect $TE = \mathbb{E}\big[Y(1) - Y(0)\big]$. This estimand belongs to the second layer, as it does not involve cross-world potential outcomes.*

*• The natural direct effect $NDE = \mathbb{E}\big[Y(1, M(0)) - Y(0, M(0))\big]$, and the natural indirect effect $NIE = \mathbb{E}\big[Y(1, M(1)) - Y(1, M(0))\big]$. These two estimands belong to the third layer, since quantities such as $Y(1, M(0))$ involve cross-world potential outcomes.*

*• The controlled direct effect $CDE(m) = \mathbb{E}[Y(1, m) - Y(0, m)]$. It belongs to the second layer, as it is identifiable under sequential ignorability assumption $(A \perp\!\!\!\perp Y(z, m) \mid X, M \perp\!\!\!\perp Y(z, m) \mid (A, X)$; or, equivalently, $(A, M) \perp\!\!\!\perp Y(z, m) \mid X)$.*

## 4    Conclusion and Discussion

In this paper, we revisited Pearl's causal hierarchy through the lens of the potential outcomes framework. Our main contribution is a systematic, probability-distribution-based classification that maps the hierarchical layers (association, intervention, counterfactuals) to increasingly complex features of potential outcomes: marginal distributions, joint distributions, and individual-level outcomes. Essentially, second-layer estimands depend only on the marginal distributions of potential outcomes, whereas third-layer estimands additionally require a coupling between these marginals, namely the joint distribution of potential outcomes. Different identification strategies can therefore be interpreted as imposing different structural assumptions on this coupling. For example, monotonicity restricts the admissible couplings, copula models parameterize the dependence structure, rank preservation corresponds to a deterministic coupling aligning quantiles, and partial identification methods characterize the set of feasible couplings consistent with the observed data.

One observation from our discussion is that estimands in the second layer have been extensively studied in the literature, whereas estimands in the third layer have received increasing attention in recent years, reflecting their importance in addressing more nuanced and complex scientific questions. Indeed, second-layer estimands (e.g., ATE and QTE) are well understood and identifiable under ignorability and overlap assumptions, whereas third-layer estimands—whether cross-world quantities (e.g., probability of causation and principal effects) or individual-level quantities (e.g., ITEs)—pose fundamentally more challenging problems. In this paper, we reviewed a range of strategies for addressing these challenges, including monotonicity assumptions, copula models, Pearl's three-step procedure, and conformal inference.

Looking ahead, we highlight several promising directions for future research, building on the perspective developed in this paper and the identification strategies reviewed herein.

*First, generalizing identification strategies for cross-world queries.* Most current identification strategies (e.g., monotonicity assumptions and copula models) are either restrictive or parametric. Developing less-restrictive and data-adaptive methods to partially or point-identify the joint distribution of potential outcomes, using tools such as optimal transport, generative modeling, or flexible bounding approaches, remains a critical research frontier.

*Second, moving from prediction to decision under third-layer objectives.* While conformal inference provides prediction intervals for ITEs, these intervals are often wide. Future research could combine conformal prediction with various partial identification assumptions to produce tighter prediction intervals, thereby yielding improved individual-level treatment rules.

*Third, algorithmic fairness and the third layer.* Our analysis of fairness metrics (Example 10) shows a clear progression from second-layer (group-level) counterfactual parity to third-layer (individual-level) counterfactual fairness. A future direction is to design machine learning algorithms that can directly optimize third-layer fairness criteria under weaker assumptions, perhaps leveraging recent advances in conformal prediction or partial identification.

In summary, we hope that the proposed potential outcomes perspective not only clarifies the existing causal hierarchy but also serves as a practical guide for researchers: helping them quickly diagnose the layer of their causal questions, assess whether their assumptions are sufficient, and navigate the growing toolkit of identification strategies. As causal inference tackles increasingly nuanced problems across diverse application scenarios, we believe that third-layer questions will only grow in importance.

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
