# OpenReview forum: "A Distributional Perspective on Pearl's Causal Hierarchy: From Marginal to Joint and Individualized Potential Outcomes"
_SLADS/Section_B — Under review for SLADS_Section_B_

### Review · Reviewer_gPTS · 2026-07-11

**Summary Of Contributions:**

This is a review-style conceptual article that aims to clarify the interpretation of Pearl’s causal hierarchy.

**Audience:**

Yes

**Broader Impact Concerns:**

Please see my comments in the Strengths And Weaknesses section

**Claims And Evidence:**

Yes

**Requested Changes:**

I have some specific comments, many of them related to claims about 3rd layer estimands.

First, some of the contributions seem to be overstated. For example, in the contributions section, the authors write “ The proposed perspective also reveals a general organizing principle underlying Pearl’s causal
hierarchy: each ascent in the hierarchy requires identifying a progressively richer probabilistic object”. This seems somewhat too strong to me. Is this not already well known?

Second, I am not sure about the claimed broad applicability of the ladder of causation discussed on page 3. Could the authors provide more concrete examples? In particular, I would like to see more practical and conceptual guidance on when the third ladder is useful or appropriate. It would be helpful to discuss this in relation to work on separable effects (and sometimes controlled direct effect), which often argues that certain third-layer estimands can be made redundant in practice. I am sympathetic to this view…

Also, the claim that “Estimands at this layer often provide complementary informatiom to those at the second layer, supporting more nuanced treatment selection and evaluation” is not convincing to me as a practical argument, at least for the first sublayer of the third layer. The authors should explain more clearly when this complementary information is genuinely useful for scientific or treatment decisions.I would also like to see a section discussing the challenges and potential limitations of third-layer estimands. This discussion could build on ideas and work by Robins and many others on interventionist mediation analysis, separable effects, and related approaches.

Finally, I would like to see a more detailed discussion of the average treatment effect on the treated, or ATT. The ATT is sometimes claimed to be third-ladder estimand, for example in work by Bareinboim (which I think fails to consider a history of relevant work by Robins and others, and I don’t believe that the ATT should be regarded as third ladder estimand, at least it doesn't need to).  This distinction deserves a more careful explanation, see, for example (there are many other works on the topic):

Bareinboim, Elias, et al. "On Pearl’s hierarchy and the foundations of causal inference." Probabilistic and causal inference: the works of judea pearl. 2022. 507-556.

and

Yang, Hongshuo, and Elias Bareinboim. "A hierarchy of graphical models for counterfactual inferences." Advances in Neural Information Processing Systems 38 (2026): 130941-130988.

**Strengths And Weaknesses:**

In my opinion the results are presented in a more sober way than in many of the naive presentations I have seen (including how Pearl presents it), and I  appreciate that. I wonder how well this type of article will age, especially because pieces of this kind will likely become easier and easier to generate with LLMs etc. However, I leave this broader question to the editors. To be clear, I am not suggesting that the authors have used AI in an inappropriate way. From my perspective, their use of such tools, if any, is entirely appropriate here. My concern is only about the general type of paper.

---

### Review · Reviewer_ReeF · 2026-07-16

**Summary Of Contributions:**

This review paper provides a potential-outcomes perspective on Pearl’s causal hierarchy. Building on this perspective, the paper systematically classifies a broad range of causal estimands, with particular emphasis on second- and third-layer estimands, explains their identifiability challenges, and reviews corresponding identification and estimation strategies. It also discusses settings involving an additional post-treatment variable.

**Audience:**

Yes

**Broader Impact Concerns:**

I do not have concern on the ethical implications of the work.

**Claims And Evidence:**

Yes

**Requested Changes:**

The authors are encouraged to address the questions, suggestions, and comments listed in the “Strengths and Weaknesses” section above. None of these proposed changes, questions, or comments is critical to securing my recommendation for acceptance. Rather, they are intended to further strengthen the clarity and overall presentation of the paper. The authors may incorporate revisions where they find them appropriate and helpful.

**Strengths And Weaknesses:**

Strengths: This is a well-written review paper. I enjoyed reading it and learned several new things. The proposed distributional perspective on Pearl’s causal hierarchy is useful, and the examples of different causal estimands, together with the corresponding identifiability challenges and identification and estimation strategies, help readers develop a broader and more comprehensive understanding of causal inference and some of its important research questions. The particular emphasis on third-layer estimands is timely and valuable.

Below are my questions, suggestions, and comments:

1. Page 2: In the boxed “Potential Outcomes Perspective,” perhaps “marginal” and “joint” could be bolded or italicized to emphasize the distinction between them.

2. Page 4: Typo: “For variable V_j” should be “For a variable V_j”.

3. Page 6: The paper states that “we omit the covariates \(X\) unless explicitly stated”. I was wondering whether all the estimands, assumptions, and identification arguments in these sections can be formulated conditionally on \(X\). If so, the authors may wish to confirm this in the paper to avoid potential confusion for readers. If not, it would be helpful to indicate which results have straightforward conditional analogues and which do not.

4. Page 7: In Example 3, it might be helpful to write explicitly
$P(Y(0)=0\mid A=1,Y=1) = P(Y(0)=0\mid A=1,Y(1)=1)$
(and similarly for PS), so that it is clearer that the joint distribution of potential outcomes is involved, as in the other examples in this section.

5. Page 8: For discrete outcomes, the no-effect probability P(Y(1)-Y(0)=0) may be nonzero. For completeness, the authors may wish to mention this quantity and briefly discuss whether it is relevant to the applications considered.

6. Page 8: Do Y^1 and Y^0 in the definition of CVaR denote Y(1) and Y(0), respectively? If so, it would be better to use consistent notation throughout.

7. Pages 8–9: It seems that the type of treatment and outcome (binary, ordered, multivalued, categorical, or continuous) affects both the difficulty of the identification problem and the proposed strategies, since the assumptions in this section depend on the treatment and outcome types. It would be helpful to clarify this point by adding some discussion.

8. Page 11: Typo: a space seems to be missing in “question,‘given the observed evidence...’”

9. Page 13: Typo: In the definition of the principal causal effect, should s1 and s0 be written with subscripts as s_1 and s_0?

10. Page 13: It would be helpful to provide some concrete examples of what S may represent in Example 8.